# Effects of Early-Life Stress on the Brain and Behaviors: Implications of Early Maternal Separation in Rodents

**DOI:** 10.3390/ijms21197212

**Published:** 2020-09-29

**Authors:** Mayumi Nishi

**Affiliations:** Department of Anatomy and Cell Biology, Nara Medical University, Kashihara 634-8521, Japan; nmayumi@naramed-u.ac.jp; Tel.: +81-744-29-8822; Fax: +81-744-29-7199

**Keywords:** neglect, c-Fos, HPA axis, reward-seeking behavior, transgeneration, epigenetics, group-housing

## Abstract

Early-life stress during the prenatal and postnatal periods affects the formation of neural networks that influence brain function throughout life. Previous studies have indicated that maternal separation (MS), a typical rodent model equivalent to early-life stress and, more specifically, to child abuse and/or neglect in humans, can modulate the hypothalamic–pituitary–adrenal (HPA) axis, affecting subsequent neuronal function and emotional behavior. However, the neural basis of the long-lasting effects of early-life stress on brain function has not been clarified. In the present review, we describe the alterations in the HPA-axis activity—focusing on serum corticosterone (CORT)—and in the end products of the HPA axis as well as on the CORT receptor in rodents. We then introduce the brain regions activated during various patterns of MS, including repeated MS and single exposure to MS at various stages before weaning, via an investigation of c-Fos expression, which is a biological marker of neuronal activity. Furthermore, we discuss the alterations in behavior and gene expression in the brains of adult mice exposed to MS. Finally, we ask whether MS repeats itself and whether intergenerational transmission of child abuse and neglect is possible.

## 1. Introduction

Exposure to stress during early life can have long-lasting effects, not only on brain function but also on cognitive and emotional development, and can increase the risk of developing stress-related psychopathology in later adulthood [1,2]. In humans, the most severe early-life adversities can be divided into three types: disruption, deprivation, and neglect and/or abuse [3,4].

Disruption involves long periods of separation from the main caregiver as a result of death, abandonment, or removal from the household. Deprivation is a condition in which the infant is reared under limited or inexistent attachment to an adult. This situation is well exemplified by foster homes, where children are sent because of parental neglect and/or sexual and psychological abuse [5]. Most animal models that have been developed to study the outcomes of mother–infant relationship disruption have used rats to replicate the various early-life adversities to which human infants may be subjected to, with good construct and face validity. These models are related to maternal neglect [6], maternal abuse [7], repeated separations [8], and deprivation of maternal care [9], and all have resulted in immediate and/or long-term alterations of the hypothalamic–pituitary–adrenal (HPA) axis. In fact, in rodent and primate models, adverse environments during the neonatal period seem to play a critical role in the establishment of the neurobiological regulation of behavior- and stress-related responsiveness [9,10,11]. In particular, the interruption of normal mother–pup interactions has been reported to induce persistent changes in the neurobiology, physiology, and emotional behavior in adult animals due to dysregulated programming of HPA axis responsiveness, which involves a cascade of central and peripheral events resulting in the release of corticosteroids from the adrenal glands [12,13].

The modern concept of “family” only consists of the parents and children (without grandparents and/or other relatives), and the parent–child relationships have changed significantly. Childhood abuse (such as sexual and/or physical abuse) and neglect may cause serious problems, including psychiatric diseases, alcoholism, and metabolic syndrome [14,15]. Thus, it is important to address if, why, and how early-life stress affects brain plasticity, causing altered emotional and cognitive functioning. Therefore, we reviewed the neural basis for the long-lasting effects of transient early-life stress on the brain. We focused on the effects of maternal separation (MS), a typical rodent model of early-life stress. We also focused on the HPA axis, brain region activation, behavioral phenotypes, gene expression, and epigenetic changes. Furthermore, we asked whether behavioral phenotypes are passed on to subsequent generations.

## 2. The MS Model

One of the most popular animal models of early-life adverse experiences to imitate human childhood abuses such as social deprivation and neglect is the separation of pups from their mother [16]. This model of early-life adverse experiences arises from the evidence that negative experiences in childhood likely cause various kinds of diseases later in life [17,18].

The consequences of the MS protocol depend on the developmental stage, duration (e.g., 15 min to 24 h), and number of days (e.g., 1 day to 21 days) of the separation experience [19,20,21,22,23,24]. Of note, many studies using the MS procedure have reported that separation of pups from their dam during a certain period of early life increases long-lasting depressive-like and/or anxiety-like behaviors in adulthood [25,26,27,28]. Basically, rodents exhibit low HPA-axis activity during postnatal days [5,29], whereas rodents exposed to early life MS show rather elevated stress responses due to higher HPA-axis activity [30,31,32,33]. Conversely, a separation for a short period of time (e.g., 15 min), called “handling”, seems to decrease depression, anxiety-like behaviors, and stress responses in adulthood [5,34]. In the case of natural rearing in rodents, the mother leaves her pups in the nest to look for food for short periods of time. Therefore, MS in the context of very short periods of time (e.g., handling) might be a more ethological parenting behavior.

Additionally, the effectiveness of MS is usually due to whether pups are isolated from each other or in a group of littermates. For instance, a previous study showed that rat pups that were isolated individually from the dam during post-natal day (PND)7 to PND11 exhibited a deterioration in barrel cortex formation, leading to defects in whisker-dependent behaviors, while pups separated in groups did not [35].

## 3. The HPA Axis

The HPA axis is an essential component of an individual’s capacity to cope with stress. In fact, HPA hyperactivity is observed in most patients with depression [36,37,38]. Upon exposure to a stressor, corticotropin-releasing hormone (CRH) and arginine vasopressin (AVP) are secreted from the paraventricular nucleus of the hypothalamus (PVN). CRH and AVP activate the anterior pituitary to induce the secretion of adrenocorticotropic hormone (ACTH), which activates the adrenal cortex to release corticosteroids—cortisol in humans and corticosterone (CORT) in rodents—which are end products of the HPA axis.

The stress hyporesponsive period (SHRP) is an exclusive “term” used in the context of the first two weeks of a rodents life, during which the HPA-axis activity is quickly relapsed [39]. The basal levels of serum CORT are rather low during the SHRP, whereas ACTH levels are higher [40]. Because higher levels of CORT adversely affect the development of the central nervous system, the control of CORT dynamics would be a potential neuroprotective mechanism against stress-induced excessive stimulation by glucocorticoid receptors (GRs) [41,42]. Furthermore, mother–pup interactions in rodents reduce HPA-axis activity, which also sustains the SHRP. In fact, MS induces hyperactivation of the HPA axis even during the SHRP [43,44]. Basal ACTH and CORT levels slowly rise if pups are separated from their mothers, peaking after several hours. MS deteriorates the SHRP, which induces the excessive release of CORT, exposing the brain to higher CORT levels. As a result, brain GRs are activated, altering brain structure and function in the developmental stages, with potential long-lasting effects on brain plasticity and behaviors later in life. In fact, McCormick and colleagues showed that higher levels of CORT induced by stress were detected in daily 1-h MS during PND2–9 [45]. On the other hand, Enthoven et al. indicated that HPA-axis activity became quickly desensitized in the context of daily MS for 8 h on PND3–5 [10]. My colleagues and myself previously reported that daily MS for 3 h on PND1–14 did not induce an increase in the basal levels of CORT on the PND14, while just a single MS for 3 h on PND14 increased the basal level of CORT [46]. Conversely, many studies have reported that daily MS for 3 h on PND1–14 significantly increases the basal CORT levels in adulthood [46,47,48]. This indicates that MS during PND1–14 could be critical for the regulation of basal CORT levels in adults.

Another study focused on CORT levels in adolescent Wistar rats after daily MS for 6 h during PND1–21 [49]. The authors showed that basal CORT levels in PND25 male rats—but not PND25 female rats—significantly increased after this type of repeated MS. This indicates that adolescent female offspring do not show rearing-dependent altered HPA-axis reactivity, whereas male offspring exhibit rearing-dependent alterations. These findings add to previous knowledge about the effects of MS in adulthood, highlighting the developmental aspects after early-life stress; however, the mechanisms underlying these alterations require further investigation.

## 4. Brain Regions Activated during MS

Examining the brain regions activated during MS is valuable for an investigation of the neural networks and behavioral phenotypes affected by early-life stress. To address this issue, one of the most convincing evaluation methods is an analysis of c-Fos expression, a product of the immediate-early gene *c-fos*, which is expressed in an activity-dependent manner [50]. Via this approach, many pieces of evidence have shown that MS stimulates various brain regions in different ways depending on the timing and duration of MS [51,52]. The current authors previously examined c-Fos expression after MS, mainly focusing on the hypothalamus and limbic brain regions, and demonstrated that repeated MS and a single incidence of MS during different developmental stages and periods showed different patterns of c-Fos expression [46,53]. C57BL/6N mice exposed to 3 h of single-occurrence MS at PND14 or PND21 showed a significant increase in c-Fos expression in many hypothalamus and limbic brain regions [46]. In contrast, C57BL/6N mice exposed to 3 h of repeated MS during PND1–PND14 showed a unique c-Fos expression pattern on PND14. Basically, repetitive homotypic stimulation does not further increase c-Fos expression due to the habituation or desensitization to chronic stimulation in adult animals, while on PND14, c-Fos expression was significantly increased in many hypothalamic and limbic brain regions after two weeks of repeated MS. On the other hand, several particular brain regions, including the arcuate nucleus (Arc), bed nucleus of the stria terminalis (BST), nucleus accumbens (NAc), dentate gyrus (DG), central nucleus of the amygdala (CeA), and medial nucleus of the amygdala (MePD) showed no significant increase in c-Fos expression compared to age-matched control animals [44,51]. These findings demonstrate that, in the Arc, BST, NAc, DG, CeA, and MePD, desensitization to repeated MS stress may be induced even on PND14. On the other hand, the c-Fos expression pattern on PND21 after repeated MS showed no significant increase compared to age-matched controls in most of the examined brain regions, except in the lateral septum and CA3.

These observations indicate that desensitization to repeated MS stress might be induced in a developmental-stage and brain-region-specific manner. Rodents in the postnatal stage show a discrepancy between CORT secretion induced by stress and the expression levels of c-Fos in the PVN. These developmental-stage dependent variations of c-Fos expression observed in the repeated-MS mice could be associated with a critical period for stress responses that is modulated by the HPA axis. During this specific period, rodents show more vulnerable responses to MS and other environmental stimulations. Particularly in mice, the first two postnatal weeks correspond to this critical period, the so-called SHRP. Thus, in early life, repeated stress is unlikely to suppress c-Fos expression. Subsequently, inappropriate activation of *c-Fos* target genes may strongly affect critical neural circuit functioning. In fact, repeated MS in PND14 mice leads to the inhibition of c-Fos upregulation in certain brain regions, including the BST, NAc, CeA, and MePD, in which specific anatomical neural connections are formed. Of note, within these regions, the extended amygdala is strongly involved in anxiety, fear, and reward regulation [54]. Hence, repeated homotypic MS may induce desensitization in the extended amygdala neurons.

Furthermore, there is a possibility that the developmental alterations in the expression of c-Fos in repeated-MS mice could be induced by glucocorticoid secretion. Of note, CORT levels are higher on PND21 than those on PND14, including in control mice, as per previous studies [55,56]. In addition, the possibility that *c-Fos* gene transcription is suppressed by the complex of glucocorticoid and GRs cannot be excluded [57]. On the other hand, c-Fos expression in PND21 mice may be suppressed due to a certain change in stress response at the neural circuit level rather than due to elevated CORT levels. In contrast, neurons in the subfornical organ that are affected by the osmotic pressure of body fluids [58] showed an increased c-Fos expression in both repeated- and single-MS mice, as compared with controls, on PND14; however, no significant changes were observed in any of the groups on PND21. This discrepancy may indicate the maturation of the physical resistance against the hyperosmolality induced by a deprivation of lactation during MS.

## 5. Alterations in Behaviors Induced by MS

The disruption of the relationship between a child and a caregiver is one of the most sensitive factors impacting lifelong mental health [5], leading to transdiagnostic features common in many psychological disorders [59]. Numerous reports have revealed alterations in the behavior of offspring exposed to MS. Here, several behavioral aspects observed in MS animal models will be summarized.

### 5.1. Depression and Anxiety Disorders

Vulnerability to emotional disorders including depression and anxiety-like behaviors derives from stressful events during early life. Adverse experiences in early life, especially interference with mother–pup relationships, have been associated with serious psychiatric diseases in adulthood [60]. As close contact with the mother is very crucial, MS in early life leads to a significant reaction of protest and despair in the pups and induces the repeated stimulation by stress mediators such as glucocorticoids and catecholamines. These disturbances of physiological and hormonal alterations result not only in unipolar and bipolar depression, but also in anxiety disorders [61,62,63,64]. Previous human studies have indicated that childhood adversity including childhood abuse and neglect leads to the prolonged and disrupted mobilization of the stress response, resulting in major depression and anxiety disorders [65,66]. These phenomena respond to anhedonic behavior and deterioration of dopaminergic reward pathways in adult animals exposed to MS in early life. MS in early life induces hyperactivity of the HPA axis that is long-lasting, even in adulthood [67,68]. In fact, many studies of repeated MS during the first two weeks in rodents caused depression and anxiety-like behaviors in adulthood [69,70,71,72,73]. In these studies, the locomotor activity and rearing behaviors decreased, while the immobility during a forced swim test and the time spent in the closed arms of an elevated plus maze increased.

A recent meta-analysis examined the effects of MS on anxiety-like behaviors such as exploratory-defensive behavior in rodents. Of note, meta-analyses are usually employed in clinical research to create evidence across various analyses, but are rarely performed in the context of animal studies [74]. The particular purposes of these meta-analyses are to measure the effect sizes of MS on exploratory-defensive behaviors using an open field test and an elevated plus maze test (the most commonly used tests to evaluate anxiety-like behaviors in rodents), as well as to determine their significance when compared to non-separated controls. Wang et al. reported three important points [75]. First, MS induces a significant increase in exploratory-defensive behaviors in rats, but not in mice. Second, the effect size of MS on these two behavioral tests in rats is almost like the anxiogenic effect of early adversity observed in humans. Third, there is significant variability among studies that needs to be addressed, such as standardizing MS protocol and considerably increasing sample sizes).

### 5.2. Fear Response

Although many studies have investigated the relationship between depression and anxiety disorders and early-life adverse experiences, experiential studies analyzing the effects of such experiences on the development of emotional learning involving fear retention and extinction are scarce. A previous study indicated that exposure to stressors or stress hormones such as CORT can encode the maturation of the fear response. Under normal laboratory rearing conditions, neonatal rats quickly forget learned fear associations and exhibit extinction learning, while adult rats express long-lasting memories of past learned-fear associations. On the other hand, if rats are exposed to stressful conditions in early life, they might show adult-like fear retention and extinction behaviors at a younger stage. Rats exposed to early MS can make an early transition from an infant relapse- resistant extinction system to an adult-like relapse-prone extinction system [76,77]. These observations demonstrate that early-life adversities including MS might impact fear learning and memory across the life span, leading to significant implications for the understanding and treatment of post-traumatic stress disorder (PTSD) and showing unusual brain responses to fear.

### 5.3. Aggressive Behaviors

In humans, three types of adverse early-life experiences in particular seem to lead to aggression and antisocial behavior: (i) a lack of strong bonds with parents or caregivers (early emotional neglect), (ii) social exclusion and loneliness (early social neglect), and (iii) exposure to stress during childhood and puberty [64]. In the case of rodents, MS has been employed to study early-life stress-induced changes in various social behaviors, including play–fight behavior (social play) and intermale aggression in juvenile, adolescent, and adult mice and rats. In fact, a previous study showed that male juvenile and adolescent offspring exposed to repeated MS spent less time in playful social interactions but showed a higher frequency of nape contacts toward the unknown age-matched play partner [16]. Furthermore, juvenile MS rats exhibited more vigorous fur pulling and less supine postures toward the play partner. These findings indicate that the behavioral patterns observed in juvenile offspring exposed to MS can be interpreted as inappropriate social play behavior, including aggressive elements. Importantly, the effects of MS experiences on social behavior continue into adulthood. For example, Veenema et al. [78] demonstrated that adult male MS rats displayed a higher level of aggression when confronted with a male intruder in their home cage in the 10-min resident-intruder test.

### 5.4. Reward-Seeking Behaviors

Using c-Fos expression as a criterion for neural activity, the current authors previously found that the extended amygdala—a brain region involved in reward-seeking behavior consisting of the CeA, MePD, BST, and NAc—was responsible for desensitization to repeated MS stress during PND 1–14 [79]. Importantly, the NAc is connected to the ventral tegmental area (VTA) via two different (direct and indirect) pathways. In the direct pathway, dopamine type-1 receptor-expressing NAc neurons directly receive dopaminergic projections from the VTA, playing a critical role in reward-associated behaviors [80,81]. We focused on this neural network and analyzed reward-seeking behaviors using a conditioned place preference test in which milk chocolate was employed as the natural reward. Adult female mice exposed to repeated MS during PND1–14 showed a significant reduction in preference score, suggesting a decrease in reward-seeking behavior for natural palatable food [79].

In contrast, offspring exposed to repeated MS exhibited a greater reward-enhancing effect after acute amphetamine administration [82]. Furthermore, ethanol consumption is affected by dopaminergic neurons in the VTA projecting to the NAc, which alters GABAergic transmission. Several studies have reported that MS increases pup vulnerability to ethanol during adolescence, inducing an increase in ethanol consumption [83,84]. These findings mirror the phenomena in humans who suffer child abuse and neglect in early life, showing higher morbidity of drug abuse and alcoholism. Taken together, these results suggest that early-life stress may increase an individual’s vulnerability to substance abuse when exposed to a drug and/or alcohol challenge in adulthood, while early-life stress may decrease an individual’s motivation for seeking natural rewards [84].

### 5.5. Behavioral Characteristics under Group-Housing Conditions

Conventional behavioral examinations in the context of MS model mice are designed to be used under simplified artificial conditions for a short observation period. Thus, the behavioral phenotypes of MS mice under more ethological conditions, such as long-term group-housing, remain largely unknown. To overcome these problems, the IntelliCage, a radiofrequency identification (RFID)-based automated behavior testing system, was employed. A previous study investigated access to the chambers as an index to analyze novel object response, behavioral flexibility, and competitive dominance with minimal experimenter intervention via the use of IntelliCage. The authors showed that MS male mice exhibited increased novelty response and were subordinate to control littermates when competing for reward access in a group-housed environment [85]. However, although IntelliCage can simultaneously analyze the behavior of over a dozen group-housed mice in the experimental cage for a long period, it is only capable of detection when a mouse comes near the RFID sensors located at the four corners of the cage; thus, it is unable to determine the exact location of the mice in the experimental cage at all times. Consequently, the behavioral phenotypes of MS mice under group-housing conditions remain largely unknown.

To investigate mouse behavior under group-housing conditions for a long period, Endo et al. recently developed a behavioral analysis system, the Multiple Animal Positioning System (MAPS), which can identify multiple mice in a group-housing environment and continuously localize the Cartesian coordinates of each mouse over long durations (Figure 1) [86]. Using the MAPS, it was found that socially reared mice affected the social proximity of isolation-reared cagemates [86]. Furthermore, the current authors have showed previously that, compared with C57BL/6J mice, BTBR mice—a model of autism spectrum disorder—exhibited altered social behavior and lower activity levels in the dark phase [87]. Importantly, the findings detected by MAPS are different from those detected by conventional open-field tests. These findings demonstrate that analyses of animal models under a more ethologically relevant condition, such as a long-term group-housing observation, should be considered and more valuable information should always be sought when attempting to understand complex behavioral phenotypes in various animal models.

## 6. Gene Expression Alterations Induced by MS

The effect of maternal separation, particularly childhood neglect, on long-lasting alterations in gene expression can be devotedly reiterated in animal models. Previous studies have indicated that variations in parenting behaviors are associated with the differential expression of genes that regulate behavioral and endocrine responses to stress [88]. As one of the neural bases of these permanently altered gene expression patterns induced by early-life adverse environments, epigenetic mechanisms (e.g., DNA methylation) affecting the chromatin structure, post-translational modifications of histones, and non-coding RNAs are advocated [89,90,91,92,93]. Meaney is a pioneer in the research of experience-dependent chromatin plasticity in association with maternal care. One of the most striking works of his group is a study of the effects of naturally occurring variations in maternal care on gene expression. Importantly, that group showed that increased maternal care such as licking and grooming activates nerve-growth-factor-inducible factor A (NGFI-A) expression through 5-HT, cAMP, and PKA, which results in the recruitment of histone acetyltransferases. As a result, histone acetylation increases, which in turn facilitates the access of demethylase and demethylation of the *GR exon I_7_* promoter in the offspring of high licking and grooming mothers. Conversely, in the absence of increased licking and grooming (and NGFI-A expression), the promoter remains methylated to decrease the expression of GR in the hippocampus, which causes higher anxiety in the offspring of low licking and grooming mothers [94].

Recently, the current authors found that early MS can reduce reward-seeking behavior in adult female mice, which intrigues the suppression of mesolimbic reward pathway. For this reason, we analyzed the dopamine receptor gene expression in NAc and found that the expression of dopamine receptor type 1 (*Drd1*) was significantly reduced in adult female mice exposed to repeated MS during their first two weeks. Furthermore, methylation patterns across *Drd1a* CpG islands significantly increased compared to age-matched control mice (Figure 2) [79]. Other recent studies have reported that histone deacetylase (HDAC) inhibition reduces VTA dopamine neuronal hyperexcitability involving AKAP150 signaling following MS in juvenile male rats [95,96]. In this paper, the authors indicated the potential use of HDAC inhibitors as a novel class of drugs for the prevention of MS-induced dopaminergic system dysfunction [97]. Furthermore, Murgatroyd et al. have showed that in the parvocellular subdivision of the PVN, MS persistently increases *Avp* gene expression associated with reduced DNA methylation of a region in the *Avp* enhancer in mice. This early-life stress-responsive region serves as a binding site for methyl-CpG binding protein 2 (MeCP2), which in turn is regulated through neuronal activity. They also found that the ability of MeCP2 to control the transcription of *Avp* and induce DNA methylation depended on the recruitment of epigenetic machinery components [98,99]. However, DNA methylation differences were also often strain-specific [100]. Taken together, these findings indicate that it is important to analyze environmental effects on a range of genetic backgrounds, strengthening the requirement for further examination of environmental, genetic, and epigenetic interactions.

## 7. Transgenerational Effects of MS

The perception that abused children grow up to become abusive parents has been broadly accepted in the field of child abuse and neglect [101,102]. However, because many other aspects of individual life (e.g., natural abilities, biological or genetic susceptibility, and intervening relationships) may reconcile the influences of child abuse and neglect, evaluating the intergenerational transmission of abuse and neglect is challenging. Some studies have supported the intergenerational transmission of abuse [103,104,105], whereas other groups have observed no evidence for transmission [106,107].

Maternal effects of the expression of defensive responses could reflect the environmental experience of the mother that is translated into the variation of the offspring phenotype via an epigenetic mechanism of inheritance. Actually, the adverse environments around pups may not only affect the development of the offspring, but also transmit its effects into the next generation [108]. Another group demonstrated that chronic and unpredictable stress during early postnatal life causes depression and alters responsivity to aversive environments in adult mice [109]. In these mice, the above-mentioned traits were partially transmitted to the subsequent generations. Transmission occurs through males and affects the offspring in a sex-dependent manner. The authors also demonstrated that early-life stress induces alterations in DNA methylation in the germline of the stressed males, with either increased or decreased methylation depending on the locus. Several candidate genes, such as those encoding for MeCP2, cannabinoid receptor 1, and CRH receptor 2, are affected. These alterations are maintained in the germline of the stressed males and are also found, in part, in the subsequent generations in both the brain and the male germline. In the case of nonhuman primates (macaques), a previous study showed that maternal abuse of offspring shares some similarities with child maltreatment in humans, including its transmission across generations [105]. This study suggested that the intergenerational transmission of infant abuse in rhesus monkeys is the consequence of early life experience and not of genetic inheritance. The availability of a nonhuman primate model of child abuse and neglect affords the opportunity not only to proceed with research on the cause-and-effect relationships of this phenomenon, but also to examine various forms of intervention, which will most certainly lead to the development of new methods for preventing child maltreatment.

In a prospective cohort clinical study, Widom et al. [102] reported that compared with individuals matched as closely as possible for age, sex, race, and approximate social class, adults who experienced childhood abuse and neglect showed an increased risk of being reported to child protective services for sexual abuse and neglect, but not for physical abuse. However, it is difficult to conclude causality from any human behavior, particularly in the natural environment, where, in contrast to the laboratory, evaluations are rather difficult. More cautious and sophisticated future studies are required to understand the mechanisms of intergenerational transmission of abuse and neglect.

## 8. Conclusions

Early-life stressful events during neonatal and postnatal periods disturb HPA-axis programming and induce long lasting effects on brain plasticity, which in turn causes various kinds of psychiatric diseases later in life. Experiments on rodent MS have shown various kinds of phenomena, including alterations in region-specific brain activation, behavioral phenotypes, and gene expression, with a prominent diversity depending on the conditions of the MS (e.g., developmental stage at testing, single-instance or repetitive). Moreover, separation conditions (e.g., duration time, isolation with or without a littermate) can also affect the obtained results. Whether MS repeats itself and whether intergenerational transmission of child abuse and neglect is possible should be clarified in future studies.

## Figures and Tables

**Figure 1 ijms-21-07212-f001:**
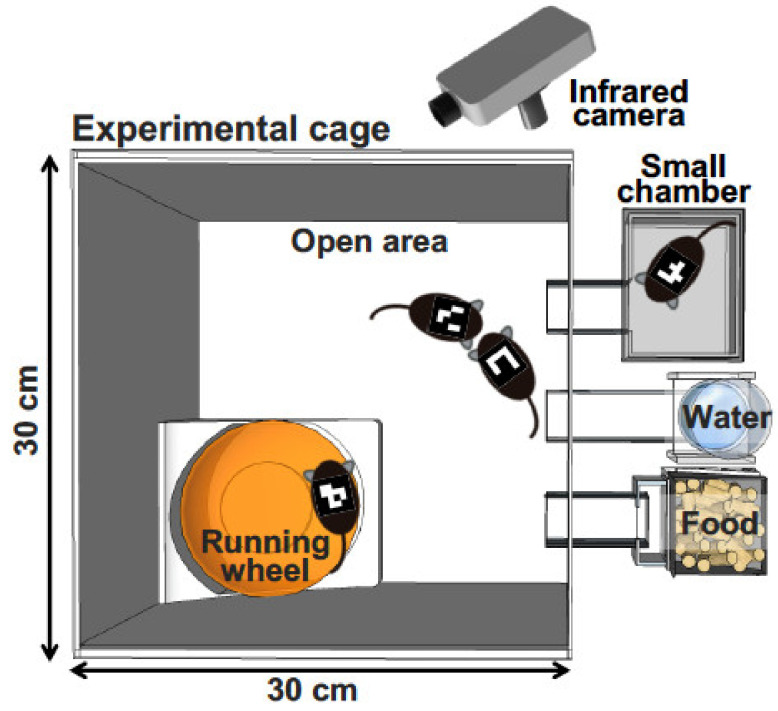
Schematic representation of behavioral analysis under group-housing conditions. Adult male mice are individually tagged with a mouse ID on their back. An infrared camera takes an image for several days. The software recognizes the ID of each mouse and records the X–Y coordinates.

**Figure 2 ijms-21-07212-f002:**
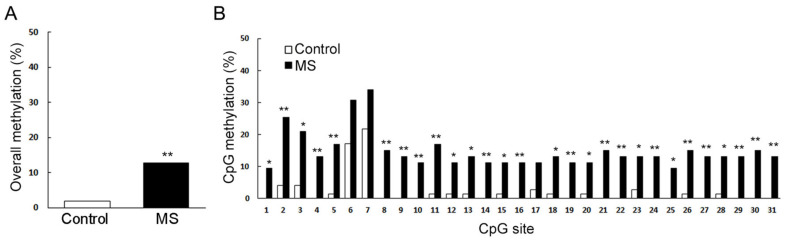
Methylation patterns across *Drd1a* CpG islands in female maternal separation (MS) mice. (**A**) Overall methylation is presented as the percentage of methylated sites (n = 9 for control, n = 8 for MS; *p* < 0.0001). (**B**) Individual CpG methylation statuses are presented as the percentage of methylated sites (*n* = 9 for control, n = 8 for MS; 31 CpG sites). * *p* < 0.05, versus control; ** *p* < 0.01, versus control (Fisher’s exact test) [61].

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
