# Peer review of "Effects of Early-Life Stress on the Brain and Behaviors: Implications of Early Maternal Separation in Rodents"

_ijms, 2020, doi:10.3390/ijms21197212_

Round 1

Reviewer 1 Report

The manuscript by Dr. Mayumi Nishi is well written and provides a comprehensive review of maternal separation as a tool for investigating the consequences of early life stress in rodents. The topics described in the review are timely and will provide useful information to those investigators interested in the downstream consequences of maternal separation on the behavioral and molecular endophenotypes associated with this type of stressor. In my notes, I had a couple of suggestions that should be described in the context of this review. In the Introduction, Dr. Nishi describes 3 types early life adversities: disruption, deprivation, and/or abuse. He then describes maternal separation as an example of disruption.  At this point, the author might want to discuss animal models of deprivation and/or abuse with appropriate references. This would allow the reader to pursue these types of ELA if interested.

In section 4, Dr. Nishi reviews the brain regions activated in response to maternal separation using cfos immunoreactivity. That is, expression of the immediate early gene, cFos, is induced in response to neuronal stimulation and increased immunoreactivity is often used as an indicator of neurons activated in response to receptor stimulation. It is well known that this phenomenon undergoes a refractory period that lasts some 16 hours or more following stimulation. It isn’t clear whether Dr. Nishi is discussing desensitization in the context of the refractory period or whether desensitization refers to desensitization of the receptors responsible for the cfos induction. It’s also p possible that he is referring to the desensitization of gene activation of cfos as something that is distinct from either of these possibilities. Some additional description would be valuable in clarifying these issues. In addition, on line 128, it should read…’in many more brain regions’…

Overall, the above issues were deemed relatively minor. This is an interesting article that will provide novel information in the context of the maternal separation literature.

Author Response

>The manuscript by Dr. Mayumi Nishi is well written and provides a comprehensive review of maternal separation as a tool for investigating the consequences of early life stress in rodents. The topics described in the review are timely and will provide useful information to those investigators interested in the downstream consequences of maternal separation on the behavioral and molecular endophenotypes associated with this type of stressor. In my notes, I had a couple of suggestions that should be described in the context of this review.

Response: We appreciate the reviewer’s comments. As indicated in the responses below, we have taken all of the comments into account during the revision of our manuscript. Please note that the changes made to the revised manuscript in response to your points are highlighted for your convenience. Please see the revised manuscript in the revised file.

>In the Introduction, Dr. Nishi describes 3 types early life adversities: disruption, deprivation, and/or abuse. He then describes maternal separation as an example of disruption.  At this point, the author might want to discuss animal models of deprivation and/or abuse with appropriate references. This would allow the reader to pursue these types of ELA if interested.

Response: We thank the reviewer for this valuable comment. Probably, my description of maternal separation was inappropriate. I do not mean maternal separation just as an example of disruption; it only includes deprivation and/or abuse. Especially, maternal separation of rodent models includes maternal neglect, maternal abuse, repeated separation, and deprivation of maternal care.

>In section 4, Dr. Nishi reviews the brain regions activated in response to maternal separation using cfos immunoreactivity. That is, expression of the immediate early gene, cFos, is induced in response to neuronal stimulation and increased immunoreactivity is often used as an indicator of neurons activated in response to receptor stimulation. It is well known that this phenomenon undergoes a refractory period that lasts some 16 hours or more following stimulation. It isn’t clear whether Dr. Nishi is discussing desensitization in the context of the refractory period or whether desensitization refers to desensitization of the receptors responsible for the cfos induction. It’s also p possible that he is referring to the desensitization of gene activation of cfos as something that is distinct from either of these possibilities. Some additional description would be valuable in clarifying these issues. In addition, on line 128, it should read…’in many more brain regions’…

Response: We appreciate the reviewer’s indication. As the reviewer pointed out, there are various possibilities for the developmental changes in the context of c-Fos expression-desensitization induced by repeated maternal separation in pre-weaned mice. It would be difficult to clearly discriminate the possibilities. Thus, I mentioned that c-Fos expression in the repeated maternal separation group showed developmental-stage and brain region-specific changes (please check lines 136-149 of the revised manuscript). Furthermore, I added some discussion about the different possibilities in section 4. Please see the highlighted sentences, in lines 150-156 of the revised manuscript.

With respect to the text in 128, I edited the manuscript to improve readability. Please check.  

>Overall, the above issues were deemed relatively minor. This is an interesting article that will provide novel information in the context of the maternal separation literature.

Response: Thank you for the positive evaluation, and for all of the pertinent raised points.

Reviewer 2 Report

This manuscript focuses on the role of maternal separation (MS) as an early life stressor and reviews the effects of MS on a number of behavioral and neural outcomes in offspring. Overall, the manuscript only moderately advances the current literature. It would be more useful if there were some novel interpretation or insight into the results of this paradigm. As it is, it doesn’t seem to present new ideas or insights. There are a number of issues with the manuscript in its current form that should to be addressed.

Line 46. It is unclear how this sentence relates to the rest of the paragraph or what the author is trying to say with it. Please clarify.

Line 53-54. “…we addressed whether MS repeats itself…”. This phrase is not needed in this sentence. It seems like the authors are simply asking whether there is transmission of a behavioral phenotype to subsequent generations. As written, it sounds like a MS experiment will be done with the next generation as well.

Section 2. The MS model. This section needs to have additional citations included that refer readers to the original work on the MS models. This model was developed and has been used for several decades, yet the citations given are all fairly recent. Please include original work be Levine and others (Levine, 1955; Levine and Lewis, 1957; etc.).

Line 64. What postnatal days do rodents show low HPA axis activity? Are these unmanipulated animals?

Line 92. “…maintained at a low level by well parental behaviors…”. What does this refer to? I think there is a typo here.

Lines 96-103. The literature cited here appear to have conflicting outcomes. Is there a possible explanation for this?

Section 4. Brain regions activated during MS. This section is very dense and hard to follow. It would be useful to have a paragraph break or two, as well as additional information about several of the findings presented. For instance, in line 121-122, the authors refer to fos expression in “mostly brain regions we observed” but it is not clear what those regions are.

Section 7. Transgenerational effects of MS. There is a body of work on the transmission of abuse of infants in rhesus macaques from the Maestripieri group. This work should be included as evidence in this section.

The English language editing is rough in a few sections. The manuscript would benefit from additional language editing.

Author Response

>This manuscript focuses on the role of maternal separation (MS) as an early life stressor and reviews the effects of MS on a number of behavioral and neural outcomes in offspring.

Response: We are grateful to reviewer #2 for the critical comments and useful suggestions that have helped us to improve our paper considerably. As indicated in the responses that follow, we have taken all of these comments and suggestions into account in the revised version of our paper. Please note that the changes made to the revised manuscript in response to your points are highlighted for your convenience. Please see the revised manuscript in the attached file.

>Overall, the manuscript only moderately advances the current literature. It would be more useful if there were some novel interpretation or insight into the results of this paradigm. As it is, it doesn’t seem to present new ideas or insights. There are a number of issues with the manuscript in its current form that should to be addressed.

Response: We appreciate the reviewer’s comment. I mostly agree that the present review “moderately” advances the current understanding of the effects of maternal separation on the brain and behavior. Importantly, I introduced a novel behavioral analysis under a more ethologically relevant condition, such as long-term group-housing with social context in section 5.5 of the revised manuscript. I believe this new system is promising and will provide more valuable information to understand complex behavioral phenotypes in various animal models in the future. Furthermore, I also discussed a new meta-analysis study examining the effects of MS on anxiety-like behavior in the section 5.1. Please check.

>Line 46. It is unclear how this sentence relates to the rest of the paragraph or what the author is trying to say with it. Please clarify.

ResponseIn this sentence, I wanted to highlight that the meaning of “family” changed. Please check the revised sentence in lines 46 and 47 of the revised manuscript.  

>Line 53-54. “…we addressed whether MS repeats itself…”. This phrase is not needed in this sentence. It seems like the authors are simply asking whether there is transmission of a behavioral phenotype to subsequent generations. As written, it sounds like a MS experiment will be done with the next generation as well.

Response: Thank you for the remark. I changed the sentence as follows: “Furthermore, we asked whether behavioral phenotypes are passed on to subsequent generations.”

>Section 2. The MS model. This section needs to have additional citations included that refer readers to the original work on the MS models. This model was developed and has been used for several decades, yet the citations given are all fairly recent. Please include original work be Levine and others (Levine, 1955; Levine and Lewis, 1957; etc.).

Response: According to the reviewer’s comments, I cited the above original works by Dr. Levine and colleagues. Please see the new references 17 and 18.

>Line 64. What postnatal days do rodents show low HPA axis activity? Are these unmanipulated animals?

Response: Rodents show low HPA axis activity mostly during the first two weeks due to the hyporesponsive period. Please find the detailed explanation in section 3 - The HPA axis.

>Line 92. “…maintained at a low level by well parental behaviors…”. What does this refer to? I think there is a typo here.

Response: I agree with the reviewer. The sentence was unclear. Therefore, I have deleted it.

>Lines 96-103. The literature cited here appear to have conflicting outcomes. Is there a possible explanation for this?

Response: One of the reasons would be due to the extremely sensitive period of the HPA axis, the stress hyporesponsive period. Another possible explanation is that these maternal separation protocols are affected by experimental conditions including the time of sample collection, the timing of maternal separation, the rearing conditions etc., which may lead to lingering questions.

>Section 4. Brain regions activated during MS. This section is very dense and hard to follow. It would be useful to have a paragraph break or two, as well as additional information about several of the findings presented. For instance, in line 121-122, the authors refer to fos expression in “mostly brain regions we observed” but it is not clear what those regions are.

Response: According to the reviewer’s suggestion, I divided this section into three paragraphs. Lines 122-123 of the revised manuscript now read:  “…in many hypothalamus and limbic brain regions.” Moreover, I also added some new information and discussion in this section. Please see the highlighted sentences.

>Section 7. Transgenerational effects of MS. There is a body of work on the transmission of abuse of infants in rhesus macaques from the Maestripieri group. This work should be included as evidence in this section.

Response: According to the reviewer’s indication, I cited the paper from the Maestripieri group (PNAS 2005; 102: 9726-9). Please see the new reference number 105 and the corresponding highlighted sentences.

Round 2

Reviewer 2 Report

The author has made addressed each of the concerns with this revision. This has significantly improved the manuscript.